

# Automatic generation of objective footprint outlines

Jens N. Lallensack

Section Paleontology, Institute of Geosciences, Rheinische Friedrich-Wilhelms-Universität Bonn, Bonn, Germany

## ABSTRACT

The objective definition of footprint margins poses a central problem in ichnology. The transition from the footprint to the surrounding sediment is often continuous, and the footprint wall complex, requiring interpolation, approximation, and a priori assumptions about trackmaker anatomy to arrive at feasible interpretations of footprint shapes. The degree of subjectivity of such interpretations is substantial, and outlines produced by separate researchers can differ greatly. As a consequence, statistical shape analysis, regardless if based on linear and angular measurements or on the shape as a whole, are neither fully repeatable nor objective. Here I present an algorithm implemented in the programming environment R that is able to generate continuous footprint outlines based on three-dimensional models—fully automatically, objectively, and repeatable. The approach, which is based on contour lines extracted from the model, traces the outline at the point where the slope of the track wall is steepest. An option for automatic landmark placement is implemented for tridactyl footprints. A case study was carried out on 13 footprints of a single trackway of a theropod trackmaker from the Lower Cretaceous of Münchehagen, Germany. Analysis of the landmark coordinates returned by the script did reproduce statistical results published in an earlier study that was based on human-made interpretative drawings, demonstrating the applicability of the present method for the objective and quantitative shape analysis of tracks. Although faint anatomical details are not always recorded and features not related to the foot anatomy may be included, the generated outlines tend to correspond with human-made interpretative drawings regarding the overall shape. While not suited as a full replacement of interpretative drawings, these generated outlines may be used as an objective basis for such interpretations.

# INTRODUCTION

Fossil footprints are an important supplement to the body fossil record, given their abundance and nature as life traces that directly record behavior and locomotion. Yet, the potential of analyses combining footprint and body fossil data is not yet exhausted, partly due to the slow advancement of objective and quantitative methodology in ichnology. A central problem in applying such methods to footprint data is the inability to objectively define the margins of a footprint, especially when the footprint indistinctly grades into the surrounding sediment. *Falkingham (2016)* demonstrated that the length of a footprint

Corresponding author
Jens N. Lallensack,
jens.lallensack@uni-bonn.de,
info@dinospuren.de

can vary as much as 27% depending on the height level chosen for measurement. Various criteria for the identification of the footprint margin have been proposed, including the point of inflexion of the footprint wall, the minimum outline, and the selection of a single contour line, amongst others (*Falkingham, 2016*). However, none of these criteria is unambiguously applicable to a wider range of different footprints, which typically show multiple inflexion points and often do not show distinct minimum outlines (*Falkingham, 2016*; *Lallensack, Van Heteren & Wings, 2016*). Adding to the problem, the vast majority of ichnological publications does not specify the criteria used for defining the footprint margins. The inability to objectively define footprint margins is highly problematic especially when quantitative methods are to be applied to analyze footprint shape, since such analyses can only generate fully objective results when based on objective input data (*Falkingham, 2016*).

The problem persists when not only linear and angular measurements but a single, two-dimensional outline abstracting the whole shape of the footprint is to be extracted. The outlines of one and the same footprint, when drawn by separate researchers, can differ considerably from each other (*Thulborn, 1990*), which repeatedly led to calls for caution in interpreting such data (e.g., *Sarjeant, 1975*; *Thulborn, 1990*; *Falkingham, 2010*; *Falkingham, 2016*). Furthermore, the high degree of simplification of two-dimensional outlines has been criticized, proposing that the full three-dimensional profile should be retained instead (e.g., *Ishigaki & Fujisaki, 1989*; *Belvedere et al., 2018*). Nevertheless, outline drawings remain the most widely used means for distributing footprint shape data, also because most anatomical information of the footprint is contained in its outline. Problematically, it is often not possible or desirable to excavate and archive footprints in museum collections, which is why material is often difficult to access or get degraded by weathering (*Bennett et al., 2013*). Ichnologists, therefore, are in many cases forced to rely on subjective outline drawings presented in the literature for ichnotaxonomic attributions and comparisons with relevant material.

Recent efforts to increase objectivity in footprint research rely on 3D-digitization techniques, most importantly photogrammetry, which allows for the fast and cost-effective capturing of footprint morphologies in high resolution (e.g., *Falkingham, 2012*; *Mallison & Wings, 2014*; *Matthews, Noble & Breithaupt, 2016*). A relatively new set of methods in the field, these techniques promise to solve critical problems of collection and dissemination of footprint data, and have been recently accepted as best practice in the documentation of fossil footprints (*Falkingham et al., 2018*). Thus, the availability of such models can be expected to further increase in the future.

Although a number of methods for the analysis of footprint shapes exist, none can effectively solve or circumvent the problem of the definition of footprint margins. Comparative approaches using 3D geometric morphometrics (e.g., *Bennett et al., 2016*; *Belvedere et al., 2018*) will include both the footprint and the surrounding sediment unless the footprint margin has been defined a priori. Therefore, such methods are feasible only when foot posture, most importantly the interdigital angles, is constant, as otherwise regions of the footprint may get averaged with surrounding sediment. Furthermore, the registration of the separate footprints still requires user-defined landmarks, which often

cannot be placed unequivocally in the absence of an objective definition of the footprint margin.

The algorithm presented herein allows for the fully objective and automatic generation of continuous outlines based on 3D surface models of footprints. The method relies on the steepness of the footprint slope (i.e., the inflexion point of the footprint wall), the probably most commonly used criterium for the definition of footprint margins (*Ishigaki & Fujisaki, 1989*).

## METHODS

The algorithm presented herein (source code provided as Data S1), implemented in the programming environment R, allows for the fully objective and automatic generation of continuous outlines based on 3D surface models of footprints. Many required functions have been already implemented in the wealth of packages available for R; these were used whenever possible, reducing the script to approximately 1200 lines of code. The implemented R function, named "oboutline", will perform the import of the 3D model and the calculation of the outline automatically in a single step, without requiring additional human input. The output of the script consists of vector plots (.svg) including the objective outline, the coordinates of the objective outline (.csv), and a .ply file of the input model fitted to the horizontal plane. When running with support for tridactyl footprints enabled, the script will additionally return landmark coordinates, a resampled version of the objective outline with homologous points, and a vector plot including the landmarks.

### Model import, orientation, and contour line generation

Import of 3D-models is achieved using the vcgImport function of the Rvcg package (*Schlager, 2017*), which supports commonly used formats including the widely used PLY. The supplied 3D-model should contain only a single complete footprint as well as a margin of surrounding surface. The exact extent of the surrounding margin will not alter results except when an increased margin size includes additional large-scale continuous contours that can be mistakenly recognized as forming part of the footprint margin. After import, the script rotates the point cloud of the extracted vertex point xyz-coordinates to fit the horizontal plane (i.e., the tracking surface on which the animal walked) by employing principal component analysis (PCA) on the three variables (x,y,z). The PCA fits three orthogonal axes (PC1–3) to the point cloud. PC1 is defined as the axis of greatest variation, with PC2 and PC3 capturing successively less variation. In most situations, PC1 and PC2 will represent the horizontal plane (the plane of greatest variation), and PC3 the relief (i.e., the deviation from the horizontal plane). Problematically, the point cloud can get mirror-inverted during PCA fitting as the signs of the columns of the rotation matrix are arbitrary, a problem also occurring in respective implementations in 3D-mesh software like Meshlab (tested with version v2016.12) or CloudCompare (tested with version 2.9). The present script calculates the Procrustes distance (a measure of shape difference) of a subset of points of the model before and after the PCA fit, and will mirror back when detecting a significant difference. If the footprint is a cast (convex hyporelief) rather than a mold (concave epirelief), it will be automatically mirrored into a mold. The scale of the input 3D

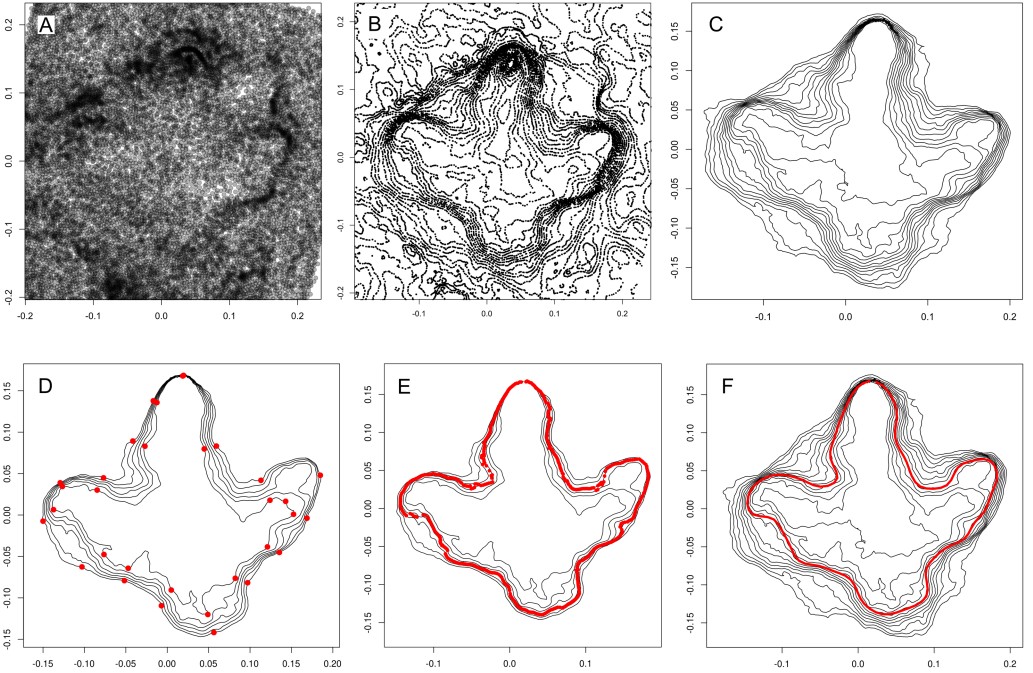

**Figure 1** **Procedure of calculating objective outlines of footprints using ornithopod footprint I1-31 from the Lower Cretaceous of Münchehagen, Germany as example.** All six steps are carried out automatically. Axes scales are in meters, and all plots are in z-direction (top view). Plots can be reproduced using the script and the 3D-model provided in File S1 and File S2. (A) The xyz point cloud is extracted from the submitted PLY mesh, and subjected to principal component analysis (PCA) for fitting to the horizontal plane. (B) Contour lines for 30 height levels are extracted based on the point cloud (xyz-coordinates of contours are shown). (C) Incomplete and short contours are removed. (D) Further contours are removed based on Procrustes distances (i.e., shape similarities). In order to establish correspondence between points of separate outlines, bottleneck points are determined along the outlines (red points). Sections in-between bottleneck points are resampled to equal numbers of equidistant points for each contour, so that each point of a contour has homologous counterparts on the other contours. (E) The location of the steepest slope is computed for each set of homologous points. Multiple slopes are taken into account by taking weighted means of the coordinates. (F) Elliptical Fourier transforms are used to fit an approximating curve to the succession of points, providing a smooth, continuous outline.

model is preserved throughout the process; measurements of the generated outline can be taken from both the plots and coordinates returned by the script.

All subsequent computations are based on contour lines of 30 equally spaced height levels extracted from the point cloud using the "getContourLines" function of the contoureR R package (*Hamilton, 2015*, Fig. 1B). Contour lines reduce the complex three-dimensional problem to a simpler and easier-to-handle two-dimensional one, and form the natural basis for 2D footprint outlines. Before the objective outline can be extracted, a number of additional steps are required, including (1) the removal of contours not representative for the footprint wall and (2) the establishment of homology between the points of separate contours.

Contours not representative for the footprint wall are excluded based on simple criteria. First, all open contours are removed, eliminating structures that continue beyond the

margins of the model. Second, only the longest contour of each height level is selected and kept, respectively, with all others removed. This results in a stack of continuous contours, with one contour per height level. Third, all contours less than 50% of the length of the longest contour are removed, while assuring that no gaps within the stack are being created. This approach eliminates smaller structures within the footprint that are unlikely to contain relevant information on the footprint wall (Fig. 1C). An option is implemented that allows for processing multiple impression per model, which is useful in cases where the footprint is not defined by a single outline. For each additional impression, the contour selecting procedure is repeated with the contours selected for the previous stacks excluded.

The resulting stack of contours may still include a number of contours that convey little or no information on the footprint wall, including roundish contours around the actual footprint. To eliminate these contours as well, and to limit the height range under consideration, areas of all contours are computed as a measure of form difference. Starting from the middle contour of the stack, the differences in area of each contour with its next lowest (or highest) neighbor are compared; if the difference in area between two contours exceeds a pre-defined threshold-value, the upper (or lower) of this contour and all following contours are removed (Fig. 1D). Different threshold-values are defined for the lower and the upper half of the stack. An option for adjusting these values is available, allowing to influence how many lower or upper contours are to be included, possibly changing the height level of the resulting calculated outline. All footprints presented in this work were calculated using the default parameters.

## Homologization of contours

Even if the starting point would correspond between all contours and if each contour would contain an equal number of equidistant points (requirements not fulfilled a priori), the individual points of the separate contours would tend to deviate from each other when far from the starting point, as the shapes of the contours are not identical. For this reason, when producing a simple mean shape, points would be averaged obliquely rather than perpendicularly to the footprint wall, leading to erroneous results. The implemented solution detects a number of "bottlenecks"—pairs of points with minimum distance between the innermost and outermost contour. Points forming the bottleneck will be considered homologous (define a line that is assumed to be perpendicular to the footprint wall), and the points in-between the bottlenecks will be interpolated by resampling.

First, all contours are resampled to the same number of equidistant points, using 500 points per default. The resulting contours can be variously oriented clockwise or counter-clockwise; contours are reversed accordingly to achieve uniform orientations. Second, Euclidean distances between all possible pairs of the inner and outer contour of the stack are calculated and stored in a matrix with the dimensions n x n. The pair with the minimum distance, the first bottleneck, is then extracted, and those points of the intermediate contours are detected that are closest to a line defined by the bottleneck points. The resulting set of homologous points is then defined as the starting point of the contours. Third, additional bottlenecks are detected to establish homology. The implemented algorithm first detects a second bottleneck on the side of the footprint opposite to the first bottleneck; two additional

bottlenecks are then found on each side between the first and second bottleneck. More bottlenecks are detected within the intermediate sections if the latter are long enough. In all cases, bottlenecks in proximity to existing bottlenecks are prevented, assuring a roughly equal distribution of bottlenecks along the outline (Fig. 1D). Finally, the individual sections between the bottlenecks are resampled to equal numbers of equidistant points, which can now be considered homologous.

## Tracing of the steepest slope

The objective outline will be traced along the steepest slope of the track wall. For each point within each set of homologous points, the minimum distance between the neighboring contours is measured. A set of homologous points is not always fully perpendicular to the footprint wall, especially when the section between the bottleneck points is long and contours differ much in orientation. For this reason, the algorithm does not simply compute the distances within the set of homologous points, but the distances between each of the homologous points and all nearby points within and outside of the set. The steepest slope computed this way (Fig. 1E) is seldom continuous along the whole outline, but rather tends to fade out and continue on a different height level, frequently leading to abrupt steps in the outline that are obviously incompatible with human interpretations. For this reason, the algorithm does not only detect the steepest point, but takes into account the steepness at all other points. Then, the final coordinate of the steepest slope is computed as the weighted arithmetic mean of all these points. Points will receive equal weight only when the steepness is equal; the lower the steepness compared to the steepest slope, the less weight is given.

The continuous and smooth final outline is produced by applying an approximating function to the resulting set of points. Of various tested options, including B-splines and Bézier curves, elliptic Fourier transforms were found to produce results most similar to those expected from a human interpreter (Figs. 1F, 2). Elliptic Fourier transforms are performed using the "efourier" function of the R package Momocs (*Bonhomme et al., 2014*) using 25 harmonics and 10 smoothing iterations.

As the resulting objective outline and contour stack will be rotated arbitrarily, an algorithm attempts to rotate both outline and stack into an upright orientation. This algorithm is based on the assumption that digit impressions are facing upwards and require a longer contour segment (i.e., more equidistant points) to be described. In a first step, the centroid, or center of mass, of the outline is computed. Then, the length of the part of the outline above the centroid is calculated for all rotation angles (1–360°), and the outline rotated according to the angle that maximizes this length.

## Automatic landmark placement

The script, as described above, can be applied to any kind of depression, as no a priori assumptions on trackmaker anatomy are introduced. Such assumptions are required when linear and angular measurements are to be extracted or when different outlines are to be aligned for quantitative shape analysis. Therefore, a function is included that returns landmark coordinates for tridactyl footprints, which are the most common dinosaur
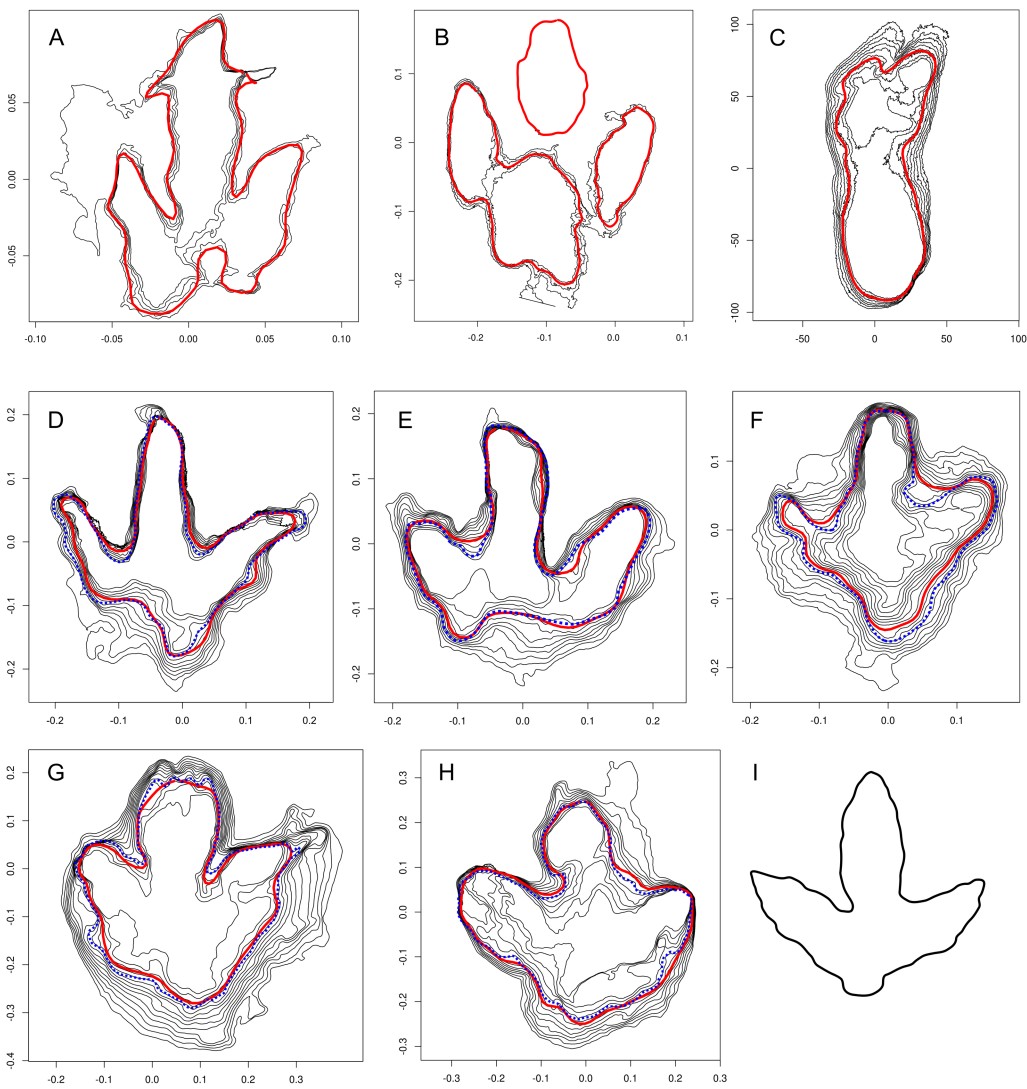

**Figure 2  Objective outlines calculated for various footprints (continuous red lines).** (A) Footprint BSY1020-E2 from the Courtedoux–Bois de Sylleux tracksite, Upper Jurassic, Switzerland (*Castanera et al., 2018*). The generated outline is affected by a crack running transversally through the central digit impression. (B) Ornithopod footprint (*Caririchnium kyoungsookimi*) from the Jindong Formation, South Korea, which is a shallow footprint consisting of three separate impressions. (C) Hominin footprint G1-35 from the Pliocene of Laetoli, Tanzania. (D–F) Footprints of theropod trackway T3 (A: T3/47; B: T3/37) and ornithopod trackway I1 (I1/35; C) from the Lower Cretaceous Münchehagen locality, Germany (*Lallensack, Van Heteren & Wings, 2016*). (G–H) Tracks 6 (G) and 5 (H) of a large tridactyl trackmaker (specimen QM F10322) from the Upper Cretaceous of Lark Quarry, Australia (*Romilio & Salisbury, 2014*). The interpretive outlines were based on a selected contour line. (I) Redrawing of a previous interpretation of track 5 by *Thulborn* (*2017*, fig. 5 (3)) d, which was drawn based on different assumptions on the trackmaker identity and markedly differs from the generated outline (H).

footprints and one of the most common footprint types in general. The algorithm first detects and separates the three digit impressions, assuming that the outline was correctly rotated in a more-or-less upright orientation in the previous step. Then, the rotation angle
of the outline is further refined by considering the central digit impression only, which is commonly considered to approximate the mid-axis of the footprint (*Leonardi et al., 1987*). A total of six landmarks are defined as in *Lallensack, Van Heteren & Wings (2016)*, on the tips of the three digit impressions, on the two hypex points, and on the heel. The landmarks on the tips of the digit impressions are defined as the distal ends of the digital axes; this definition reduces the influence of claw impressions, which may vary strongly depending on preservation and behavior (*Lallensack, Van Heteren & Wings, 2016*). The digital axes were computed by rotating the respective digit into the upright position and taking the mean of the x-coordinates; the intersection with the outline was found using the rgeos R package (*Bivand & Rundel, 2018*). The two hypex points were computed by finding the lowest point between the enclosing digit impressions relative to a line connecting the landmarks at the tips of these digit impressions. The landmark on the heel region is the intersection of the axis of the central digit impression and the proximal margin of the outline.

## Case study and sensitivity analysis

The applicability of the presented approach for the quantitative analysis of footprints was tested on 13 footprints pertaining of a single theropod trackway (T3) from the Lower Cretaceous of Münchehagen, Germany (3D-models are provided as data S2). Geometric morphometric analysis of the same footprints was conducted in earlier studies based on interpretive drawings (*Lallensack, Van Heteren & Wings, 2016*; *Wings, Lallensack & Mallison, 2016*), suggesting that (1) the landmarks on the hypex positions and on the heel are more variable than the landmarks on the digital tips, and that (2) the lateral hypex is more variable than the medial hypex. In the present case study, the calculated objective outlines were aligned using Generalized Procrustes Analysis (GPA) with the geomorph R package (*Adams, Collyer & Kaliontzopoulou, 2018*). The mean shape of the aligned shapes was then compared with the meanshape published by *Lallensack, Van Heteren & Wings (2016)*. This is expected to reveal potential systematic differences between the traditional interpretive approach and the automated approach presented herein. Furthermore, landmarks are automatically extracted from all 13 generated outlines as described above, and their variability in y-direction (parallel to the footprint mid-axis) computed, in the same way as has been done in *Lallensack, Van Heteren & Wings (2016)* based on interpretive outlines.

## RESULTS AND DISCUSSION

### Limitations

Human-made interpretational footprint drawings aim to capture as much information about the trackmaker anatomy as preserved. Although slope steepness is the most important criterion, the steepest slope will frequently fade out along the footprint wall to continue at a different height level, making interpolation unavoidable. Furthermore, humans tend to make a priori assumptions about trackmaker anatomy when producing the outlines, which allows them to take into account extramorphological (unrelated to the foot anatomy)

features and include anatomical features of interest such as digital pad impressions and claw marks.

The present algorithm is successful in detecting and interpolating outlines even when the steepest parts of the slope are indistinct (Fig. 2). It does currently not include a priori assumptions that would emphasize anatomically important details and account for extramorphological features, which keeps the algorithm simple and predictable, and applicable to a wide range of footprint types. The broad applicability is demonstrated in Fig. 2C with a hominin footprint from the famous Laetoli tracksite of Tanzania (*Leakey & Hay, 1979*). In the absence of a priori assumptions, however, the outlines expectedly tend to provide less information on the presumed foot anatomy than interpretative drawings, and may be unusable in cases where track morphology is obscured by extramorphological features. Artifacts caused by a crack running transversally through the central digit impression are shown in Fig. 2A for a medium-sized theropod footprint from the Late Jurassic Courtedoux–Bois de Sylleux tracksite of Switzerland (*Castanera et al., 2018*).

Additional limitations currently arise from the necessity to limit the vertical extent of the contour stack, which may exclude relevant anatomical features not captured by the selected contours. Furthermore, the described approach requires that impressions can be described by single contours, which is not the case in all cases, especially when the footprint is very shallow and indistinct. However, a preliminary option is implemented to process footprints that are composed of more than one impression, as demonstrated with an ornithopod footprint attributed to *Caririchnium kyoungsookimi* from the Jindong Formation in Goseong County, South Korea (Figs. 2B, 2D data provided by Anthony Romilio). This footprint is relatively shallow (maximum depth is 3.8% of maximum length of model) and comprises three separate impressions.

## Qualitative comparisons with human-made outlines

The similarity of the generated outlines with human-made interpretations is demonstrated on five fossil footprints that have been previously published in the literature (Figs. 2D–2I). All five examples are compared with published interpretative drawings that had been produced using the same 3D-models; all examples, as is the case for examples used elsewhere in this work, were processed using the same script version and parameters. Footprints T3/47, T3/37, and I1-35 (Figs. 2D–2F, respectively) come from the Lower Cretaceous of Münchehagen, Germany (*Lallensack, Van Heteren & Wings, 2016*; *Wings, Lallensack & Mallison, 2016*). All three footprints were left by the right foot. T3/47 and T3/37 were part of a larger theropod and I1/35 (Fig. 2F) of an ornithopod trackway. The objective outlines (red continuous lines) are generally in accordance with the interpretational drawings published in *Lallensack, Van Heteren & Wings (2016)* (dotted blue lines). However, the sediment bars between the digital impressions tend to be less extensive (e.g., Figs. 2E–2F), and digital impression IV in T3/47 is somewhat abbreviated in the objective outline due to sediment infilling in the distal tip of the impression. The generally good match between the generated outlines presented herein and the interpretive ones of *Lallensack, Van Heteren & Wings (2016)* may be partly due to the use of similar criteria for the definition of the track

margin, which are discussed in detail in the latter study, most importantly the criterion of the steepest slope.

Tracks 5 and 6 of QM F10322 (Figs. 2G–2H) are the left and right pedal impressions of a large tridactyl trackway from the Upper Cretaceous of Lark Quarry in Queensland, Australia (*Thulborn & Wade, 1984*; *Romilio & Salisbury, 2011*; *Romilio & Salisbury, 2014*). The 3D-models, based on photographs of the in situ specimen taken in 2013, were provided by Anthony Romilio. The trackway became famous after it was suggested to have been left by a large theropod causing a dinosaur stampede (*Thulborn & Wade, 1984*), a hypothesis that has been discussed controversially in recent years (*Romilio & Salisbury, 2011*; *Romilio & Salisbury, 2014*; *Thulborn, 2013*; *Thulborn, 2017*; *Romilio, Tucker & Salisbury, 2013*), with the identification of the large tridactyl trackmaker as either a theropod or an ornithopod constituting a major point of disagreement. This discussion is instructive in showing how much interpretative outlines can differ when produced by separate researchers with different preconceptions about the responsible trackmaker taxon. Well aware of the subjectivity problem, *Romilio & Salisbury* (*2014*, fig. 7F, 8F) did not produce traditional outline drawings but selected a single contour line they considered representative. Still, this approach is not completely objective, as separate contours can differ greatly in shape and dimensions (*Falkingham, 2016*). The calculated outlines presented herein correspond well with those of *Romilio & Salisbury (2014)* (Figs. 2G–2H). However, they differ considerably from outlines of the same imprints presented by *Thulborn & Wade* (*1984*, plate 17), and especially from a more recent interpretation of track 5 by *Thulborn* (*2017*, fig. 5 (3)), which is redrawn here for comparison (Fig. 2I). Track 5 (Figs. 2H–2I) is a prime example of how different interpretational drawings can be when based on fundamentally different assumptions. Given their interpretational nature, it is, on principle, not possible to discard one of these disparate interpretations as incorrect (*Falkingham, 2016*). Furthermore, the good correspondence of the generated outline with the outline presented by *Romilio & Salisbury (2014)* could be partly due to the fact that both outlines were produced based on contour lines, while the differing outline of *Thulborn (2017)* was traced on a photograph. Nevertheless, the presented approach may offer a standard to which interpretive outlines can be compared. It furthermore allows for an objective qualitative comparison of different footprints, as generated outlines would be free of preconceived assumptions on the trackmaker taxon and computed using the same parameters. Although not a replacement for interpretive outlines given the limitations outlined above, generated outlines may be used in combination with interpretive ones, and may form the objective basis for the production of the latter.

## Uses in quantitative shape analysis

All 13 footprints of the T3 trackway included in the original geometric morphometric analysis of *Lallensack, Van Heteren & Wings (2016)*, *Wings, Lallensack & Mallison (2016)* were processed using the same program version and parameters (Fig. S3), and analyzed following the protocol provided by the mentioned studies. The mean shape derived from the Procrustes-aligned generated outlines closely matches the previously published mean shape that was based on interpretive outlines of the same tracks (Fig. 3C). This indicates

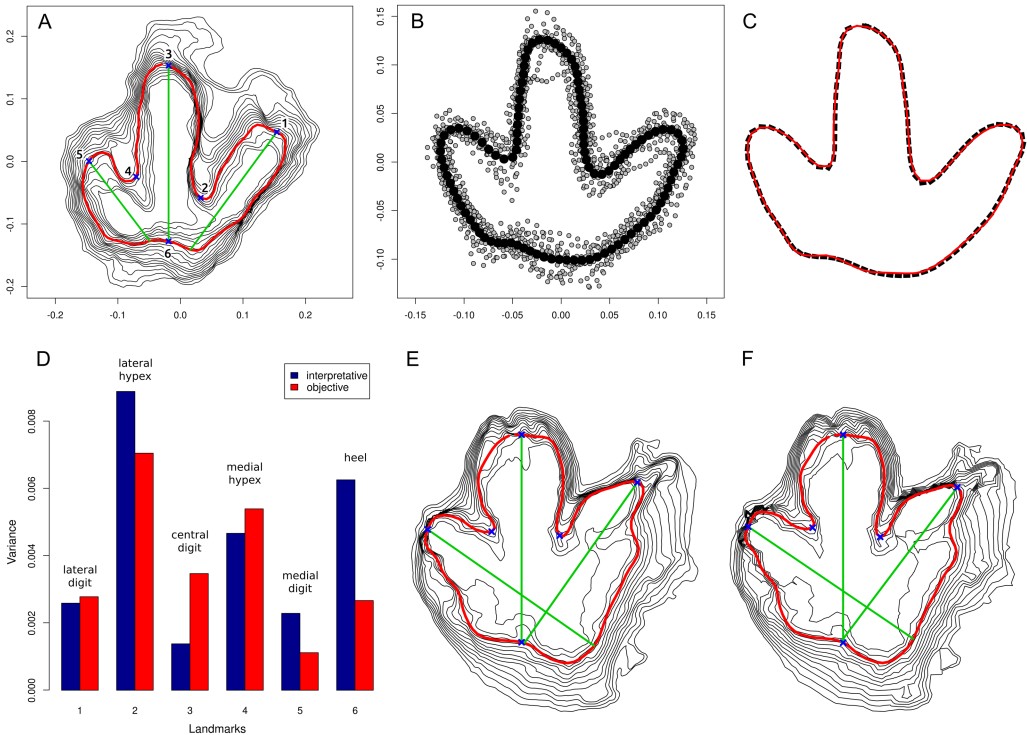

**Figure 3** **Quantitative evaluation of the present approach.** (A) Footprint T3/23 from Münchehagen, Germany, processed with option to automatically place landmarks enabled. The six generated landmark points are shown as blue crosses; computed digital axes are shown as green lines. (B) Procrustes analysis of 13 footprints from trackway T3 of Münchehagen. Black dots represent the mean shape, and grey dots the individual aligned outlines. (C) Comparison of the mean shape automatically generated by the script based on the objective outlines (black line, dotted) and that published by *Lallensack, Van Heteren & Wings (2016)* based on interpretive outlines (red line, solid). (D) Comparison of the variability of landmarks, with those derived from the interpretive outlines in blue (left columns) and those produced by the present script in red (right columns). (E–F) Sensitivity analysis, comparing the script output of the full-resolution model (E; 196,236 faces, 9.4 MiB) with that of a model of reduced size (F; 5000 faces, 246.2 KiB) of track 6 of QM F10322 from Lark Quarry, Australia.

that, at least in this case, systematic differences in generated and interpretive outlines are minor. Analysis of the variability of the six landmarks (as indicated in Fig. 3A), however, reveals more substantial differences (Fig. 3D). The objective approach confirms that the lateral and medial hypex (landmarks 2 and 4) are more variable than the remaining landmarks. However, the previously published observation that the heel is highly variable is not supported. Furthermore, the central digit (landmark 3) is markedly more variable according to the objective approach. This is partly due to sediment infilling of the distal end of this digit impression (an extramorphological feature) in one of the footprints, which is not taken into account by the objective approach (Fig. 3B). This case study demonstrates that generated outlines can be used to objectively reproduce results derived from interpretive outlines, although the possible influence of extramorphological features and other potential errors in the objective outlines need to be taken into account.

To test for repeatability of results generated by the present approach, one model (track 6 of QM F10322) was saved at successively lower resolutions using MeshLab, and the resulting outlines were compared (Figs. 3E–3F). Observed differences resulting from the separate model resolutions are found to be negligible, even when the highest resolution (Fig. 3E; 196,236 faces, 9.4 MiB) is compared with the lowest tested resolution (Fig. 3F; 5,000 faces, 246.2 KiB).

## CONCLUSIONS

The lack of widely applicable, objective means for defining the footprint margin is among the most vexing problems in the research of fossil footprints. The present algorithm automatically generates continuous objective footprint outlines by employing the criterium of the steepest slope. In contrast to human-made interpretive outlines, these generated outlines allow for analyses that are fully reproducible and free of interpretational bias, as complete samples can be processed using the same mathematically defined criteria. Although the generated outlines tend to correspond well with interpretive outlines, extramorphological features unrelated to the foot anatomy may be incorporated, and anatomical detail not captured by the steepest slope may be excluded. While not a fully appropriate replacement for human-made drawings in most cases, computed outlines may be used as an objective basis for the production of interpretational drawings, and allow for objective qualitative comparisons. Most importantly, the approach paves the way for fully objective quantitative analyses of footprint shapes.

## ACKNOWLEDGEMENTS

I would like to thank Michael Buchwitz, Karl Bernhardi and Vincent Bonhomme for discussions, ideas, and help with writing the presented software. Anthony Romilio and Matthew Bennett provided additional 3D models that were important in testing this software.

### Funding

The author received no funding for this work.

### Competing Interests

The author declares there are no competing interests.

### Author Contributions

- Jens N. Lallensack conceived and designed the experiments, performed the experiments, analyzed the data, contributed reagents/materials/analysis tools, prepared figures and/or tables, authored or reviewed drafts of the paper, approved the final draft.

## Data Availability

The raw data is available from the following repositories:

Romilio, Anthony (2019): Photogrammetric 3D models of tracks 5 and 6 of Lark Quarry trackway QM F10322 and of Caririchnium kyoungsookimi from South Korea. figshare. Dataset.

https://doi.org/10.6084/m9.figshare.7718549.v1

Belvedere, Matteo; Castanera, Diego; Marty, Daniel; Paratte, Géraldine; Cattin, Marielle; Lovis, Christel; et al. (2018): A walk in the maze: Variation in Late Jurassic tridactyl dinosaur tracks from the Swiss Jura Mountains (NW Switzerland). 3D photogrammetric models. figshare. Dataset.

https://doi.org/10.6084/m9.figshare.5662306.v2

Hominin footprint G1-35 from the Pliocene of Laetoli, Tanzania. Data from: Bennett MR. 2013. Bournemouth Fossil Footprint Archive. Available at http://footprints.bournemouth.ac.uk/.

## Supplemental Information

Supplemental information for this article can be found online at http://dx.doi.org/10.7717/peerj.7203#supplemental-information.

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
