# Peer review of "Automatic generation of objective footprint outlines"

_PeerJ, doi:10.7717/peerj.7203_

## Round 0.1 · original submission · Major Revisions

This manuscript presents an automated technique for more objective and reproducible outlines of individual fossil tracks. The reviewers are generally quite positive on this intent (and I agree), although they provide some suggestions for improvement during revision. Most specifically, some additional validation of the method should be included. Although I have indicated major revisions, in my editorial opinion this manuscript is somewhere between minor and major revisions. The method itself as presented seems OK, but its outputs must be investigated in more detail.

Specifically:

- Both reviewers (in slightly different wordings) request some more concrete demonstration of the differences (or similarities) between your automatically generated outlines and previous human-generated outlines. Figure 2 provides this to a qualitative degree, but are there some other measures you could use to compare these (e.g., differences in track outine area or length)? What is the percentage difference between (for instance) track length measured by a physical observation using calipers on the track, and track length measured from the outline? In many cases, I recognize that "important details" are subjective on outline shape, so perhaps you could also more explicitly discuss cases where directly human-generated outlines show features or highlight details that aren't in the digitally generated outlines? Some examples might be the gaps between the toes in the track in Figure 2E, or the foot arch in the track in Figure 2F. In general, I think you do a good job of emphasizing that some details will be omitted in these auto-generated outlines, but an additional repetition or two of this in the text probably wouldn't hurt. My general sense is that this should be used with caution for identifying relevant features for ichnotaxonomy, but would be quite useful for morphometric analyses.

- As suggested by the reviewers, some additional discussion and perhaps illustration of "edge cases" (no pun intended) would be helpful. In tracks where the track outline is hard to delineate (as often occurs in undertracks, for instance), what is the result?

- Data are not provided for all models (e.g., dinosaur QM F10322; hominin G1-33). Are these data archived somewhere that is publicly accessible? This is (in my understanding) necessary for reproducibility requirements per journal policy.

Some other questions to consider in revision (from Farke as the editor):

- Does the method work the same for impression (concave) vs. natural cast (convex)?
- Will the R script be uploaded to another software archive? Is it planned to be uploaded to R-CRAN (or is this the type of thing that would be accepted there?)?
- I would strongly advise including specific examples of limitations in the methods.
- The examples here are for pretty "simple" tracks. How would it hold up to things like mammalian tracks (e.g., cats, dogs, horse, camel, etc.)? Peter Falkingham (https://peterfalkingham.com/resources/) has some freely available 3D data that might be helpful to test this.
- How does the method hold up to things like cracks, irregularities in the surface surrounding the track, etc.?
- What input resolution is required? I would suggest some sort of sensitivity analysis with a model of varying resolution, to see how robust your method is to this metric of input data quality. This could be accomplished by downsampling a single model through several iterations, and perhaps comparing a specific parameter.
- Can scale be preserved in this method? I.e., how would you create measurements from the outlines? Based on the figures, it seems that you would just take it from the axis of the plot, but this should be specifically stated.
- Please double check caption for Figure 2; the lettering is off between the caption and figure. Also, please cite the sources for the dotted blue lines in Figure 2. Can you figure the multiple past interpretations for the large tridactyl trackmaker (not just the Romilio & Salisbury ones--this might have to be split into a separate figure)? What sorts of features are missed? Why?
-- What parameters can be tweaked, and how, for edge cases? E.g., arguably different settings could highlight different features, as in the gaps between toes for the track figured in 2E.

·

Basic reporting

The manuscript is unambigous and well structured and well written; literature is properly cited, and some additions should be considered only after my comments.

Experimental design

The research questions are well defined, and the code seems solid. However, the test is based only on a reduced sample size (6 footprints) and more data should be included to valdiate the quality of the approach and of the algorithms. It would be also interesting to see how the algorithm deals with steep and low/gentle walls and if the "steepest slope" approach applies properly to different kind of tracks.

The method is properly described, but need a larger sample to be validated. It also need more clear ojectives (e.g., outline comparisons) the results lacks important information for other studies (e.g., biomechanical, ichnotaxonomical)
I have added some link to published 3D material in the pdf and I'm happy to share unpublished data if you need more.

Validity of the findings

The algorithm seems solid, but need more test to be validated and to highlight the limits of the "steepest slope" approach.

Additional comments

Having tryied different objective apporach to the study of tracks I am a bit skeptical it would ever be possible to achieve a fully objective study of footprints. In any case, I think the manuscript is an important contribution to the discipline and is accetable for publication after major revisions.

The mansucript is well written, but lacks a bit of focus on the objectives: tracing an authomatic outline of the tracks, as currently possible with your code, allows broad morphological comparisons, good to study track variability, but the lack of internal details and the misinterpretation of extramorphologies prevent any ichnotaxonomical comparison or study. Such fucus should be more clearly stated in the discussion/conclusion highlighting the limits of the method.

All my comments in the attached pdf.

Reviewer 2 ·

Basic reporting

No comment.

Experimental design

No comment on the experimental design.

I recommend that demonstrating the necessity of a new method of determining track outlines, especially given how close the algorithm outlines were to the human-interpreted outlines, will strengthen the manuscript.

Validity of the findings

I would like to see more discussion of how congruent the algorithm outlines are with the human interpreted outlines. It appears that the algorithm results actually provides support for the accuracy of human interpretations and that we can use the older track outlines (especially if that's the only specimen we have to work with) with some confidence.

This leads to my next question: exactly how is this method an improvement over the conventional human-interpreted track outlines? I wish to see presented specific scenarios in which this method would be relied on and be an improvement for interpretations in ichnology. For example, would this method allow for the analysis of intra-trackway variability more efficiently than with using human-interpreted outlines? Does this method assist with interpreting very poorly preserved footprints? Also, when conducting quantitative morphometric studies using footprint outlines, is there a significant difference between the results produced using human-interpreted outlines and the results using the algorithm presented herein?

Based on the 3D models in the supplementary information, there did not appear to be a great deal of preservational variation in the test tracks. It would be beneficial to the reader to see this method applied to tracks with a variety of preservational states. The algorithm-outlines are already demonstrated to not interpret extramorphological features: what other preservational conditions might also produce suboptimal algorithm outlines? These issues should be addressed or at least discussed in greater detail.

Additional comments

The algorithm-based track outlines do have the potential to be used in morphometric studies. However, given the similarities of the algorithm-outlines to the human-interpreted outlines, the paper should clearly demonstrate that the algorithm-based outlines will have a noticeable benefit to ichnological interpretations both for morphometric analysis and for qualitative interpretations. What preservational scenarios will the ichnologist find the algorithm-based outlines preferable to traditional outlines? More discussion is needed to justify the benefit of the algorithm-based outlines over the human-interpreted outlines.

---

## Round 0.2 · accepted · Accept

Thank you for your thorough and close attention to the comments on the previous version of the manuscript. It has been improved dramatically, and is now ready for acceptance. This will be a useful contribution for many researchers!

# ·

Basic reporting

no comment

Experimental design

no comment

Validity of the findings

no comment

Additional comments

Dear Author,
I am pleased to see that all the suggestion were taken into account. The manuscript has greatly improved, and is now suitable for publication.

I still see some limitations for ichnotaxonomical purposes, as internal morphology is not recorded, but it is an important milestone to begin with. I'm confident that further development in this and other computational tools will lead to an integrated (traditional and computational) approach to ichnotaxonomy.

All the best